# Do Brief Mindfulness Interventions (BMI) and Health Enhancement Programs (HEP) Improve Sleep in Patients in Hemodialysis with Depression and Anxiety?

**DOI:** 10.3390/healthcare9111410

**Published:** 2021-10-21

**Authors:** Paola Lavin, Rim Nazar, Marouane Nassim, Helen Noble, Elizaveta Solomonova, Elena Dikaios, Marta Novak, Istvan Mucsi, Emilie Trinh, Angela Potes, Ahsan Alam, Rita S. Suri, Zoe Thomas, Clare Mc Veigh, Mark Lipman, Susana Torres-Platas, Outi Linnaranta, Soham Rej

**Affiliations:** 1McGill Meditation and Mind-Body Medicine Research Clinic and Geri-PARTy Research Group, Lady Davis Research Institute and Jewish General Hospital, McGill University, Montreal, QC H3T 1E2, Canada; rim_nazar@hotmail.com (R.N.); marouane.nassim@mail.mcgill.ca (M.N.); elena.dikaios@mail.mcgill.ca (E.D.); gabriela.torresplatas@mail.mcgill.ca (S.T.-P.); soham.rej@mcgill.ca (S.R.); 2School of Nursing and Midwifery, Queen’s University Belfast, Belfast BT9 7BL, UK; helen.noble@qub.ac.uk (H.N.); clare.mcveigh@qub.ac.uk (C.M.V.); 3Department of Psychiatry, McGill University, Montreal, QC H3A 1A1, Canada; liza.solomonova@gmail.com (E.S.); angela.potesholguin@mail.mcgill.ca (A.P.); ahsan.alam@mcgill.ca (A.A.); zoe.thomas@mcgill.ca (Z.T.); outi.linnaranta@thl.fi (O.L.); 4Psychiatry Centre for Mental Health, University Health Network, Toronto, ON M5S 1A1, Canada; Marta.Novak@uhn.ca; 5Department of Psychiatry, University of Toronto, Toronto, ON M5S 1A1, Canada; 6Multiorgan Transplant Program, Division of Nephrology, University of Toronto, Toronto, ON M5S 1A1, Canada; istvan.mucsi@utoronto.ca; 7Research Institute of the McGill University Health Center, Department of Nephrology, McGill University, Montreal, QC H3A 0G4, Canada; emilie.trinh@mcgill.ca (E.T.); rita.suri@mcgill.ca (R.S.S.); mark.lipman@mcgill.ca (M.L.); 8Department of Nephrology, Jewish General Hospital, Montreal, QC H3T 1E2, Canada; 9The Finnish Institute for Health and Welfare, 00271 Helsinki, Finland

**Keywords:** mindfulness, hemodialysis, sleep, actigraphy, meditation, psychosocial intervention

## Abstract

(1) Objective: to determine if a brief mindfulness intervention (BMI) and a health education program (HEP) could improve measures of insomnia in patients undergoing hemodialysis. (2) Methods: this was a planned secondary analysis of a randomized controlled trial of BMI vs. HEP for hemodialysis patients with depression and/or anxiety symptoms. The primary outcome for the analysis was the Athens Insomnia Scale (AIS). The secondary outcome was consolidation of daily inactivity (ConDI), an actigraphy measure that describes sleep continuity and is based on a sleep detection algorithm validated by polysomnography. We also explored whether changes in AIS and ConDI were associated with changes in depression, anxiety, and quality of life scores over 8-week follow-up. (3) Results: BMI and HEP groups did not differ significantly from one another. Exposure to BMI or HEP improved sleep quality (baseline AIS 9.9 (±5.0) vs. 8-week follow-up 6.4 (±3.9), (V = 155.5, *p* = 0.015)), but not ConDI. Improvements in AIS were associated with lower depression scores (Rho = 0.57, *p* = 0.01) and higher quality-of-life scores (Rho = 0.46, *p* = 0.04). (4) Conclusions: mindfulness and HEP may be helpful interventions to improve self-reported sleep quality in patients undergoing hemodialysis. Decreases in insomnia scores were associated with decreased depression symptoms and increased quality of life scores.

## 1. Introduction

Sleep disorders are common in dialysis patients with a mean prevalence of 60% [1], while the prevalence observed in the general population is 4% to 29% [2]. In hemodialysis patients, insomnia may lead to other complications including higher mortality [3] and lower quality of life [4]. Sleep problems are often attributed to a large body mass index, inflammatory status, low nutritional indices, comorbid anxiety or depression, inadequate dialysis, and overnight rostral fluid shift [5]. Pharmaceutical treatments such as sedative antidepressants and anxiolytics remain common interventions; however, concerns exist about habituation and complications such as drowsiness, sedation, confusion, and increased risk of falls [6,7]. Thus, non-pharmacological strategies such as promotion of sleep hygiene, exercise and cognitive and psychological approaches should be considered [1].

Research over the past two decades widely supports that mindfulness meditation exerts beneficial effects on physical and mental health [8,9,10,11,12]. Mindfulness denotes a state of consciousness that is characterized by an intentional and non-judgmental awareness of present experiences, rather than attempts to alter current experience or to eliminate them from awareness. Multiple studies have suggested that mindfulness meditation may be effective in treating some aspects of sleep disturbance [13,14,15]. A meta-analysis suggested that its practice may facilitate telomerase enzyme functioning, which is associated with health and mortality [16]. The psychological process model of sleep postulates that mindfulness can influence sleep by altering psychological factors related to sleep (e.g., awareness, decentering, acceptance, readiness to change, and motivation) [13]. A meta-analysis on mindfulness-based treatments for insomnia reported a reduction of symptoms and higher sleep quality compared to psychological placebos and waitlist controls [14]. However, previous mindfulness-based interventions for patients receiving hemodialysis have used large group trainings, long individual sessions or a combination of these, which poses a limitation for everyday implementation. Our alternative delivery design within an RCT explored the effectiveness of brief interventions across 8 weeks, assisted by simple audiovisual technology to address the resource-intensiveness of traditional group and individual mindfulness training, usually limited within the context of clinical care.

We performed a secondary analysis of an 8-week follow-up randomized controlled trial (RCT) of Brief Mindfulness Intervention (BMI) vs. Health Enhancement Program (HEP) for patients undergoing hemodialysis treatment who experience depression/anxiety symptoms. Our primary hypothesis was that BMI or HEP would decrease insomnia scores, as measured by Athens Insomnia Scale (AIS) and consolidation of sleep (ConDI) by actigraphy. Our secondary hypothesis was that BMI would have a superior effect on sleep quality compared to HEP. Our exploratory hypothesis was that lower insomnia scores would be associated with less severe depression and anxiety, and higher scores of quality of life.

## 2. Materials and Methods

### 2.1. Study Design

The primary study was an 8-week RCT of BMI vs. HEP for patients undergoing hemodialysis treatment with symptoms of depression and/or anxiety. In that study, we recruited 55 patients from the Centre Hospitalier Universitaire de Montréal, Jewish General Hospital and McGill University Health Center (May 2017–June 2018). The study was carried out in accordance with The Declaration of Helsinki of 1975, revised in 2013 [17]. Participants provided written consent, and ethics approval was obtained by all sites. Further details can be found in the parent study [18].

### 2.2. Participants

Inclusion criteria for the current study were being a patient undergoing hemodialysis treatment with depression and/or anxiety symptoms (Patient Health Questionnaire PHQ-9 score > 6 and/or General Anxiety Disorder-7 (GAD-7) > 10). Exclusion criteria were suggested dementia (Mini-Cog > 3), psychosis or acute suicidal ideation clinically assessed during the initial interview. 

### 2.3. Randomization, Concealment, Blinding, and Reducing Expectancy Bias

Randomization was stratified by site by an independent statistician. Data collectors and outcome assessors were blinded to allocation. Expectancy bias was lowered by advertising the study as “Alternative Treatments for Depression and Anxiety”.

### 2.4. Intervention: Brief Mindfulness Intervention (BMI) and Active Control: Health Enhancement Program (HEP)

While undergoing hemodialysis, participants received two chair-side 20 min sessions of BMI per week for 8 weeks. Sessions included 15 min of guided mindfulness meditation techniques drawn from mindfulness-based cognitive therapy (MBCT) [19]. These mindfulness meditation techniques included a body scan, mindful eating, guided breath meditation, mindful movement, and loving-kindness meditation. Techniques emphasized paying attention to specific elements of one’s moment-to-moment sensory experience with a non-judgmental attitude. In addition, participants learned material about mindfulness and how to apply it to daily life. Participants could interact with the interventionist for 3–5 min after each session. Interventions were delivered in English or French via audio headsets. A 10 min daily home mindfulness practice was encouraged. One interventionist was a registered social worker with facilitator certification in MBCT and a personal mindfulness practice of over 7 years. The other was a psychologist and certified Mindfulness Based Stress Reduction (MBSR) teacher and a MBCT facilitator with over 40 years of clinical mental health experience. Both the intervention and active control programs were delivered and reviewed by the same interventionist at any given site to control for the effect of interventionist characteristics and ensure consistency.

Active Control: Health Enhancement Program (HEP) has been used as an active control in mindfulness-based intervention trials to control for several non-program-specific intervention factors including facilitator attention, expectation for positive change, treatment duration, format (e.g., individual vs. group), and time spent on at-home practice [19]. It was structurally equivalent to the mindfulness meditation program (two 15 min sessions per week for 8-weeks, delivered via audio headsets in groups of 4–6 participants, with 3–5 min for questions or discussion), and encouraged the same amount of home practice (implementing health-enhancing habits for 10 min daily). Each session involved educational and activity-based sessions on light exercise, sleep, stress and anxiety, nutrition, journaling, and music enjoyment with drawing [18].

### 2.5. Actigraphy Device (GENEActiv)

Actigraphy is a procedure that records the occurrence of wrist movements over time through a wearable device similar to a watch which is used in a natural setting (e.g., everyday life). Actigraphy has been validated vs. polysomnography [20] and is able to detect wakefulness and sleep. Participants wore actigraphy devices across 24 h during ≥10 days at baseline and again ≥10 days at 8-week follow-up.

### 2.6. Outcome Measures: Assessed at Baseline and Post-Intervention

The primary outcome measure was sleep quality measured by the Athens Insomnia Scale (AIS-8) [21] at baseline and 8-week follow-up. AIS measures problems with sleep initiation, night awakening, total sleep, and sleep quality (insomnia ≥ 6). AIS has shown excellent characteristics for clinical and research use (external validity: r = 0.90, *p* < 0.001; diagnostic validity: only 1% of those responders with insomnia are misdiagnosed (i.e., NPV = 99%) and (PPV = 41%), sensitivity (93%) and specificity (85%); internal consistency: α = 0.89, test retest (n = 194; within one week): r = 0.89, *p* < 0.001) [21].

The secondary outcome measure was change in consolidation of sleep, measured by a novel actigraphy measure (ConDI) [22], developed using a validated sleep detection algorithm [23] to process the data and detect sleep continuity. The exploratory outcomes were the association of AIS and ConDI scores with depression, anxiety, and quality of life (PHQ-9, GAD-7, and EQ-5D) [24,25,26]. In our randomized sample, the groups for BMI and HEP presented differences in clinical history; however, baseline subjective and objective sleep measures showed no significant differences (AIS U = 37.5, *p* = 0.72 and ConDI U = 35, *p* = 0.58, Mann–Whitney test).

### 2.7. Statistical Methods

Normality of the data was assessed with the Shapiro–Wilk test. Baseline characteristics were described using independent T, Mann–Whitney U, or a chi-square test. We used Spearman’s correlation to analyze the association of exposure to MBI and HEP and (1) changes in AIS and (2) changes in ConDI, as well as (3) changes in PHQ-9, GAD-7, and quality of life. We compared AIS and ConDI measures between baseline and 8-weeks using Wilcoxon signed-rank test. Comparison of interventions was analyzed with the Kruskal–Wallis test and size effects with “R-statistic” and Eta-squared using R statistic software [27].

## 3. Results

The original RCT screened 112 participants (PHQ-9, GAD-7) of whom 64 met eligibility criteria for the trial. Of these, 9 were not interested in participating and 55 were randomized. There were four participants per arm who reported to feel too tired and withdrew before the first follow-up. Patients who refused the sleep assessment reported feeling too tired during the data collection (n = 33). Aiming to be as inclusive as possible, we considered the patients’ tolerance and shortened the assessment. Priority was given to collecting PHQ9 and GAD7 (outcomes of the main study). A subset of 22 patients accepted to continue assessments and provide baseline sleep data, from which three dropped-out before the follow-up. 

In our sample (n = 19) the mean age was 68.7 (±6.4), and 47% were female. Patients in both groups did not differ on baseline demographic and current clinical characteristics. The groups showed differences in clinical history; however, baseline sleep measures were comparable (Appendix A). For our primary objective, exposure to BMI or HEP was associated with an improvement in AIS scores (mean baseline AIS 9.9 (±5.0) vs. 8-week follow-up AIS 6.4 (±3.9), V = 155.5, *p* = 0.015, Wilcoxon signed-rank test). For our secondary objective, neither of these interventions were significantly associated with changes in ConDI (median baseline consolidation, 0.51 ± 0.11, 8-week follow-up 0.55 (±0.15), V = 69, *p* = 0.31 r = 0.24 (Table 1)). When comparing BMI to HEP, there were no significant differences in AIS scores (H = 13.1, *p* = 0.51, effect size: η^2^ = 0.059) or consolidation of sleep (H = 8.32, *p* = 0.87, effect size η^2^ = 0.002).

We explored whether change in AIS and change in ConDI were associated with changes in depression (PHQ-9), anxiety (GAD-7) and quality of life at 8-week follow-up. We found that decreases in AIS scores were associated with less severe depression (Rho = 0.57, *p* = 0.01) and improvements in quality of life (Rho = 0.46, *p* = 0.04), but was not significantly associated with anxiety. ConDI was not associated with changes in depression, anxiety, nor quality of life (Appendix A).

## 4. Discussion

BMI or HEP was associated with clinically important reductions in self-reported insomnia over 8-week follow-up. We did not see a difference in insomnia scores between BMI and HEP exposure, likely because HEP is a strong active control, perhaps comparable to structured psychoeducation programs, with effects that could be compared to the ones observed when using cognitive behavioral therapy (CBT) [28]. HEP’s therapeutically valid elements (e.g., healthy eating, exercise, music/art therapy) likely contributed in addition to non-specific control factors. As reported in a meta-analysis of RCTs [14], the overall effect of mindfulness has been to decrease self-reported insomnia scores. Recent findings [29,30] have suggested that awareness and acceptance could be the mechanisms of mindfulness interventions in improving sleep quality, partly via reducing psychological stress. A metacognitive model of insomnia [31] has proposed that increasing awareness of the mental and physical states that are present when experiencing insomnia symptoms and then learning how to shift mental processes can promote an adaptive stance to one’s response to these symptoms. These metacognitive processes are characterized by balanced appraisals, cognitive flexibility, equanimity, and re-commitment to values and are posited to reduce sleep-related arousal, leading to remission from insomnia. 

Reliable objective approaches for the detection and characterization of sleep problems may turn out to be relevant for clinical screening of sleep. The values from the commercially available actigraphy processing software have usually been developed in healthy subjects with regular sleep patterns and are suboptimal to alert about aberrant sleep patterns in a given patient, such as the chronically ill [22]. We used the algorithm validated with polysomnography by van Hees et al. [23] and further modified it by adding a rolling window to determine the nighttime sleep bout. We attempted this new measure for sleep continuity (ConDI) and did not observe a correlation with improvements reported by AIS with BMI or HEP. Further development of the actigraphy measure is needed for this population.

In our sample, reductions in scores of self-reported insomnia had moderate-to-strong correlations with reductions in symptoms of depression and higher quality-of-life scores. While in this sample we are not aware which component was the main or first benefactor of the intervention, the results suggest the potential use of BMI and HEP as a clinical strategy for patients undergoing hemodialysis treatment. Substantial epidemiological evidence supports that sleep disturbance (i.e., insomnia, poor sleep quality, and/or insufficient sleep) contributes to inflammatory disease risk, and that sleep disturbance and inflammation are both thought to have a role in depression [32]. Interactions between sleep and inflammation mechanisms underscore the implications of sleep disturbance for inflammatory disease risk and suggest the need to address sleep as a way to modulate inflammation, depression symptoms and promote health.

Future studies might benefit from hybrid designs of online audio-visual delivery of interventions to reduce costs and improve accessibility and scalability [29]. Virtual mindfulness sessions may be complemented with in-person check-ins, which may be an important treatment factor for psychosocial interventions.

### Strengths and Limitations

Strengths of this study include the use of an active control group and the convenience of the intervention delivery system (audio headsets and chairside interactions during hemodialysis), which allowed for easier access to treatment for patients with limited mobility and energy. Our study was limited by (1) a small sample size, (2) the lack of control for bias introduced by participants’ differences in detailed clinical and laboratory status, and (3) the lack of measure of dispositional mindfulness.

Future studies might benefit from the use of hybrid designs of online audio-visual delivery of interventions expand scalability, accessibility, and lower costs.

## 5. Conclusions

Brief Mindfulness Intervention and Health Education Programs may be helpful interventions to improve insomnia in hemodialysis patients experiencing depression and/or anxiety symptoms. Improvements in insomnia scores appear to be associated with improvements in depression symptoms and quality of life. The limitations encountered with using additional pharmacotherapy in a population that already struggles with polypharmacy and the resource intensiveness of traditional individual psychotherapy might be overcome by BMI and HEP, which may be effective, scalable, and cost-efficient alternatives. Furthermore, their implementation by virtual/online methods might offer an interesting avenue to explore interventions for insomnia.

## Figures and Tables

**Table 1 healthcare-09-01410-t001:** Effects of Brief Mindfulness Intervention (BMI) and Health Enhancement Program (HEP) on Athens Insomnia Scale (AIS) and consolidation of sleep (ConDI) (n = 19).

(A) BMI and HEP				
Sleep measure	Baseline	8-week follow-up	Statistics (Wilcoxon signed-rank test)	Effect Size (R statistic)
Athens Insomnia scale	9.9 (±5.0)	6.4 (±3.9)	V = 155.5, *p* = 0.015	r = 0.559
Consolidation of sleep	0.51 (±0.11)	0.55 (±0.15)	V = 69, *p* = 0.31	r = 0.24
**(B) BMI vs. HEP**				
Sleep measure	HEP (n = 7)	BMI (n = 12)	Statistics (Kruskal–Wallis Test)	Effect Size (Eta squared)
AIS	Pre 9.57 (±2.9), post 7.4 (±4.2)	Pre 10.1 (±6.0) post 5.8 (±3.7)	H = 13.1, *p* = 0.51	η^2^ = 0.059
Consolidation of sleep	Pre 0.57 (±0.06), post 0.54 (±0.04)	Pre 0.48 (±0.13) post 0.55 (±0.2)	H = 8.32, *p* = 0.87	η^2^ = 0.002

## Data Availability

Individual participant data that underlie the results reported in this article, after deidentification, will be available immediately after publication with no end date, with investigators whose proposed use of the data has been approved by an independent review committee identified for this purpose. Proposals should be directed to soham.rej@mcgill.ca. To gain access, data requestors will need to sign a data access agreement.

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
