# Peer review of "Do Brief Mindfulness Interventions (BMI) and Health Enhancement Programs (HEP) Improve Sleep in Patients in Hemodialysis with Depression and Anxiety?"

_healthcare, 2021, doi:10.3390/healthcare9111410_

Round 1

Reviewer 1 Report

Review: Do Brief Mindfulness Intervention (BMI) and Health Enhancement Program (HEP) improve sleep in patients in hemodialysis with depression and anxiety?

The study is designed to examine the effectiveness of BMI, compared to HEP, in enhancing sleep disturbance for patients in hemodialysis. The parent study examined the effect of BMI vs HEP in reducing depression and anxiety. The following is the questions and suggestions.

Introduction

  1. I believe that authors failed to articulate the unique contribution of this study to the research field. Authors mentioned that “mindfulness-based treatments have not yet been examined in patients receiving hemodialysis.” However, a simple google search showed some studies that investigated the effect of mindfulness-based treatment on sleep issues in patients receiving hemodialysis. Nejad et al.’s study was recognized in the parent study. Moreover, there is a review article including studies testing the same or similar research question as the current study.

Nejad, M. M., Shahgholian, N., & Samouei, R. (2018). The effect of mindfulness program on general health of patients undergoing hemodialysis. Journal of Education and Health Probmotion, 7, 74. doi:10.4103/jehp.jehp_132_17

Natale, P., Ruospo, M., Saglimbene, V. M., Palmer, S. C., & Strippoli, G. F. M. (2019). Interventions for improving sleep quality in people with chronic kidney disease. Cochrane Database of Systematic Reviews, 5, CD012625. doi:10.1002/14651858.CD012625.pub2.

Second, I am not sure that sleep disturbance can be the outcome variable enough to create one study or article. Sleep disturbance, not sleep disorder being assessed by a diagnostic tool, can be treated as one of the symptoms in depression. Supporting that sleep disturbance is a part of depression, the main outcome variable in the parent study was depression of which measurement included sleep disturbance item. Also, I believe that the importance of the sleep disturbance as the single and independent outcome variable in this study was blurred by testing the relationship between the enhanced sleep disturbance and reduced depression, which seemed redundant to the parent study.

  1. The secondary hypothesis of the study was about the effectiveness of BMI vs HEP in enhancing sleep quality. Authors expected that the effectiveness of BMI would be larger than that of HEP, but I cannot understand why (authors did not provide any background information about this comparison). Also, considering that there are many studies in the effects of psychological interventions in sleep disturbances of the patients receiving dialysis, I believe this secondary hypothesis should have been the main hypothesis.

Materials and Methods

  1. Authors could have provided more explanations of outcome measures, especially the Athens Insomnia Scale (AIS). The number of items (I believe there are at least two versions of AIS), interpretation, and psychometric properties of AIS were not included.
  2. I expected a simple analysis to test the significance of changes of AIS and ConDI in the study participants and the difference in the changes of AIS and ConDI between MBI and HEP groups, using repeated measures t tests or regression after controlling for the baseline data. I cannot understand why authors used the statistical analyses for ranking data.

Results

  1. If I understood correctly, authors changed the assessment in the middle of the study. Do all participants receive the same kind of assessment? How does the changed assessment influence the reliability of the study?
  2. According to Appendix A, two groups (BMI and HEP groups) look different in terms of history of depression and anxiety. I believe this might have influenced in the sleep disturbance in groups. Related to this difference, results did not include the difference test of baseline AIS and ConDI.
  3. The second hypothesis was about the effectiveness of BMI vs HEP in enhancing sleep quality. The result was provided in Table 1, but the results was not mentioned in the text.

Discussion

  1. I believe that the importance or implication of the significant result in the effectiveness of BMI and HEP in reducing sleep disturbance was not sufficiently mentioned.
  2. The references about the limitations in ConDI should have been included.

Reviewer 2 Report

Thank you for providing a chance to review your manuscript. Overall, the paper is well-written, but it is better to make minor revisions. Overall, this is an interesting study and provides a better understanding of mindfulness and sleep quality.

  1. Methods section

    Because Montreal is in the French-speaking part of Canada. Have all the measures you used been validited for montrealer populations? Do you use a French version or you translated them into French? If not, evidence of its validity and reliability should be provided according to the data of the analyzed sample (e.g., CFA, McDonald's omega).

  1. Discussion section

Further discussion and interpretation of the effect of mindfulness on sleep quality is necessary. It would be better to clarify the mechanism in more depth.

  1. Reference

The reference section is too old, it is better to cite more references in the last 3 years.

Reviewer 3 Report

I enjoyed the reading and do not hesitate to congratulate the authors for conducting this interesting research. 

To be cited by "mindfulners", who are willing to conduct meta-analyses and systematic reviews on the efficacy of MBI with RCT assignment, it is mandatory to point out who was trained and where they received (received) the MBI training and how much this MBI training lasted. 

Just to clarify this issue because the difference in the efficacy of different MBI research is often due to the mindfulness trainers. 

On the other hand, I understand the limitation in word count but it is sometimes advisable to indicate if this study had mediating or moderating effects between treatments and outcomes. Dee so absolutely nothing has been commented. 

Finally, it is very complicated to understand why participants did not fill a measure of dispositional mindfulness. 
